# Current Status and Application of Metformin for Prostate Cancer: A Comprehensive Review

**DOI:** 10.3390/ijms21228540

**Published:** 2020-11-12

**Authors:** Hyun Kyu Ahn, Young Hwa Lee, Kyo Chul Koo

**Affiliations:** Department of Urology, Gangnam Severance Hospital, Yonsei University College of Medicine, Seoul 06229, Korea; wharang11co@yuhs.ac (H.K.A.); wjungyh@naver.com (Y.H.L.)

**Keywords:** metformin, prostatic neoplasm

## Abstract

Metformin, an oral biguanide used for first-line treatment of type 2 diabetes mellitus, has attracted attention for its anti-proliferative and anti-cancer effects in several solid tumors, including prostate cancer (PCa). Liver kinase B1 (LKB1) and adenosine monophosphate-activated protein kinase (AMPK) activation, inhibition of the mammalian target of rapamycin (mTOR) activity and protein synthesis, induction of apoptosis and autophagy by p53 and p21, and decreased blood insulin level have been suggested as direct anti-cancer mechanisms of metformin. Research has shown that PCa development and progression are associated with metabolic syndrome and its components. Therefore, reduction in the risk of PCa and improvement in survival in metformin users may be the results of the direct anti-cancer mechanisms of the drug or the secondary effects from improvement of metabolic syndrome. In contrast, some research has suggested that there is no association between metformin use and PCa incidence or survival. In this comprehensive review, we summarize updated evidence on the relationship between metformin use and oncological effects in patients with PCa. We also highlight ongoing clinical trials evaluating metformin as an adjuvant therapy in novel drug combinations in various disease settings.

## 1. Introduction

Metformin is an oral biguanide for first-line treatment of type 2 diabetes mellitus (T2DM). Metformin decreases liver glucose production and increases insulin sensitivity and use of glucose by peripheral tissues [1]. In non-diabetic patients, metformin is used for polycystic ovary syndrome to manage metabolic disorders associated with insulin resistance [2]. Moreover, metformin’s efficacy as a potential antiviral agent is under investigation [1].

Metformin has been shown to have anti-cancer effects in various hormone-sensitive tumors, such as breast cancer, pancreatic cancer, colon cancer, and prostate cancer (PCa) [1,3]. High levels of insulin and c-peptide have been reported to be associated with poor survival outcomes in patients diagnosed with cancer [4,5]. The anti-cancer effects of metformin have been postulated to be associated with low insulin level subsequent to inhibition of hepatic gluconeogenesis [6]. In a systematic review, use of metformin in patients with T2DM has been associated with a reduction in risk of any cancer and cancer-related mortality [1]. Moreover, a meta-analysis of 20 publications, including studies of patients with T2DM and cancers, showed that use of metformin provided increased overall survival (OS) and cancer-specific survival (CSS) compared to use of other T2DM agents [3]. At the same time, the oncological efficacy of metformin and its degree of survival benefit are reported to be dependent upon cancer type. Studies have reported that metformin increases the survival of patients with breast, colorectal, ovarian, and endometrial cancers. However, in general, use of metformin has provided suboptimal results for lung cancer, pancreatic cancer, and PCa [7,8].

PCa develops in an androgen-dependent state, and studies have demonstrated that metabolic syndrome is associated with increased risk of development and progression of PCa [9,10]. Therefore, metformin has been considered a potential anti-PCa agent and has been a topic of several population-based epidemiological studies. However, while some studies have shown reduced risks of PCa incidence and mortality with use of metformin, other studies have failed to demonstrate significant benefit. In this comprehensive review, we summarize updated evidence on the relationship between metformin use and oncological outcomes in patients with PCa. We also review clinical trials evaluating metformin as an adjuvant therapy in novel drug combinations in various disease settings.

## 2. Association between Prostate Cancer and Metabolic Syndrome

For decades, evidence has suggested that metabolic syndrome and its components are associated with increased development and progression of aggressive PCa [11,12,13,14,15,16,17]. For incidental PCa, studies have reported no or an inverse relationship with metabolic syndrome [18,19]. Recently, Hammarsten et al. reported that metabolic syndrome and its components may hinder the diagnosis of low-stage PCa by a mechanism that reduces serum prostate-specific antigen (PSA) level [17]. Thus, the observed inverse relationship between metabolic syndrome and low-stage PCa may be the result of diagnostic bias, rather than the underlying biology associated with its development.

Androgen deprivation therapy (ADT) plays an important role in the treatment of advanced PCa, either as monotherapy or combined with radical prostatectomy (RP) or radiation therapy (RT). ADT may be utilized in all stages of PCa and confers increased survival in advanced stages of the disease. However, ADT is associated with a wide range of adverse effects and reduced quality of life. In terms of metabolic syndrome, reduced level of circulating testosterone induces increased circulating insulin level, insulin resistance, changes in body composition, fatigue, sexual dysfunction, decreased bone mineral density, hyperlipidemia, and acute coronary syndrome. These side effects may compromise OS outcomes [20,21].

Insulin promotes local androgen synthesis by PCa cells; this is considered one of the mechanisms in the development of castration-resistance [22]. In a retrospective study, Flanagan et al. reported that metabolic syndrome was associated with a shorter time to PSA progression and inferior OS in patients with PCa receiving ADT [23]. Therefore, if metformin can exert positive effects on hyperinsulinemia and metabolic syndrome, it may be potentially utilized as an adjunctive treatment in reducing the risk of castration resistance in patients on long-term ADT.

## 3. Mechanism of Action of Metformin on Anti-Cancer Effects

A variety of anti-cancer mechanisms for metformin has been reported to date. Two theories, in particular, have attracted attention. The first is a direct anti-cancer mechanism that is unrelated to insulin, and the second is an indirect mechanism associated with insulin (Figure 1).

### 3.1. Direct Effect

The direct anti-cancer mechanism of metformin is associated with activation of adenosine monophosphate-activated protein kinase (AMPK) and inhibition of the mammalian target of rapamycin (mTOR) activity [24]. AMPK is an energy-sensing/signaling protein central to maintaining energy homeostasis [25]. AMPK is activated by cellular stress, such as glucose deprivation, hypoxia, and oxidative stress, induced by increase in the ratio of AMP to adenosine triphosphate (ATP) [26].

Metformin enters complex 1 of the electron transfer chain (ETC) and blocks its activity, reducing oxygen consumption and ATP production. As a result, the AMP to ATP ratio is increased, and cells are stressed, which activates AMPK [27,28]. AMPK activation promotes tuberous sclerosis complex (TSC2) and inhibits mTOR activity [29]. The decrease in mTOR activity reduces the levels of 4E-binding protein (4E-BP) and ribosome protein S6 kinase (S6K) factors. This reduces protein synthesis and proliferation and subsequently inhibits cancer cell growth and proliferation [24,28,30,31].

Studies have reported activation of ataxia-telangiectasia mutated (ATM) and liver kinase B1 (LKB1) as alternative anti-cancer mechanisms of metformin [28]. ATM and LKB1 are both tumor suppressors, essential proteins that control and regulate the cell cycle. The activity of LKB1 leads to expression of the p53 gene by regulating the AMPK activity in the cell and induces apoptosis or autophagy of cells [26]. Moreover, p53 gene activation leads to rapid transcription of the p21 gene, a tumor inhibitor that reduces the proliferation of cancer cells and inhibits cell cycle progression [26]. ATM phosphorylates LKB1 and activates AMPK in response to metformin entry into the cell.

### 3.2. Indirect Effect

The indirect mechanism involves activation of AMPK by metformin. This prevents transcription of genes associated with glycogenesis in liver cells [32,33]. In this process, reduction of glycogenesis increases glucose uptake in muscle cells, thereby reducing serum glucose and insulin levels [32,33]. Insulin receptors in cancer cells in association with high insulin level increase mitogenic effects, tumor growth, and proliferation. Therefore, reduction in insulin level prevents cancer cells from proliferating [34,35,36].

The role of insulin in tumorigenesis is associated with the insulin receptor (IR), insulin-like growth factor receptor (IGF-R), and insulin-like growth factor (IGF). The insulin and IGF-1 receptors (IGF-1R) are homologous and often overexpressed in cancer cells [37]. When insulin, IGF-1, or IGF-2 binds to IR, the signal is transmitted to the cells by autophosphorylation. Subsequently, the PI3K/AKT/mTOR signaling pathway is activated, leading to abnormal cell proliferation, inhibition of apoptosis, and carcinogenesis [38]. According to several studies, IGF-1R is limited to caveolae, which are abundant in caveolin proteins that regulate vesicular transport, endocytosis, and cell signaling. Tyrosine phosphorylates caveolin 1, which is the most common caveolar protein, and has been shown to affect the anti-proliferative action of metformin [39,40].

Metformin also engages in glucose metabolism in cancer cells [41]. Cancer cells exploit various mechanisms to produce energy, including increased metabolism, nutrient demand, and glucose consumption, known as the Warburg effect, to compensate for rapid cell growth and proliferation, primarily by glycolysis [42]. Metformin suppresses glucose uptake in the tumor. The inhibitory action of metformin on glucose metabolism derives from the combination of glycolysis and its effect on the growth factor signaling pathway [41]. Hexokinase (HK)-II is an enzyme that promotes glucose phosphorylation in the mitochondrial membrane and protects cells from cell death. Metformin inhibits HK-II, which catalyzes glucose phosphorylation, thereby reducing cellular energy availability and glucose uptake to promote cell death [41]. In lung cancer, metformin has been shown to inhibit cellular glucose uptake and phosphorylation by impairing the enzyme functions of HK-I and HK-II in Calu-1 cells [43].

### 3.3. Biologic Effects

The androgen receptor (AR) has a crucial role in PCa and regulates multiple cascades of events, such as proliferation, invasion, differentiation, and apoptosis [44]. ADT is the mainstay of treatment for advanced PCa; however, a substantial proportion of patients on long-term ADT progress to castration-resistant PCa (CRPC). Persistent AR signaling is a crucial component in disease progression of CRPC, and studies have proposed AR down-regulation as a feasible treatment strategy [45]. Metformin has been reported to reduce the level of AR protein in a dose-dependent manner in AR-positive cell lines and to suppress the AR signaling pathway via down-regulation of AR mRNA. This observation supports the role of metformin as a potential adjunctive therapy to ADT [46,47].

## 4. Effect of Metformin on Prostate Cancer Incidence

Large-scale observational studies have found inverse associations between metformin use and colon cancer, liver cancer, and lung cancer. However, the results of epidemiological studies regarding use of metformin in patients with PCa have been inconsistent (Table 1). Several studies have reported a significant inverse relationship between metformin use and risk of PCa development, but other studies have failed to demonstrate any association [48,49,50,51,52,53,54,55,56,57].

The association between metformin use and PCa incidence was analyzed using data from 12,226 patients with PCa and controls in the Danish Cancer Registry and the Aarhus University Prescription Database [48]. Metformin users showed reduced risk of PCa diagnosis compared to non-users (adjusted odds ratio (aOR): 0.84; 95% confidence interval (CI): 0.74–0.96). Moreover, higher intensity of metformin use (calculated as the number of pills per day, in quartiles) and higher cumulative dose were associated with decreased incidences of PCa. However, other oral hypoglycemics did not reduce PCa risk (aOR: 0.98; 95% CI: 0.86–1.10). These data suggest the existence of a unique mechanism of metformin in suppressing PCa development [48].

Haring et al. estimated the relationship between metformin use and PCa risk using data from 78,615 patients included in the Finish National Prescription Database. In their study, use of any anti-diabetic agent was associated with a lower risk of overall PCa (hazard ratio (HR): 0.85; 95% CI: 0.79–0.92), but the risk of metastatic PCa increased (HR: 1.44; 95% CI: 1.09–1.91). Subgroup analysis revealed that metformin decreased PCa risk (HR: 0.81; 95% CI: 0.69–0.95) in a dose-dependent manner and that sulphonylureas increased the risk of metastatic PCa (HR: 2.04; 95% CI: 1.11–3.77). This observation supports the notion that hyperinsulinemia may increase the risk of PCa since sulphonylureas stimulate insulin secretion [49].

In a Dutch population, Ruiter R et al. investigated the association between metformin use and cancer risk using 2.5 million institutional records. Use of metformin, with a sulfonylurea derivative as a reference, was associated with lower risks of overall cancer (HR: 0.90; 95% CI: 0.88–0.91) and PCa (HR: 0.92; 95% CI: 0.88–0.97). Whether the reduction in cancer risk was the result of the beneficial effect of metformin or increased cancer risk with sulfonylurea derivatives is yet to be determined. Moreover, when the average doses for metformin and sulfonylurea derivatives were calculated in terms of average daily dose, dose–response relations were shown for metformin users, but not for sulfonylurea derivative users [50].

Wang CP et al. performed a retrospective study using 76,733 veterans with T2DM and reported that Hispanics and African-American (AA) men not using metformin were at higher risk of PCa incidence compared to non-Hispanic White (NHW) men not using metformin. Metformin alone or combined with statins was associated with a greater reduction in PCa incidence in Hispanic men compared to NHW men; however, this association was not found between AA and NHW men. Furthermore, combined use of metformin and finasteride was associated with a greater reduction in PCa incidence in Hispanics and AA men compared with NHW men. The authors suggested that use of metformin could be a potential strategy in preventing PCa development in minority populations that have a high risk for aggressive PCa [53].

A population-based, observational cohort study involved 2652 elderly men with preexisting diabetes and incidental PCa [54]. Metformin users were at significantly lower risk of advanced PCa compared to non-users (4.7% vs. 6.7%; *p* < 0.03). After adjustment, metformin was associated with a 32% reduced risk of PCa (aOR: 0.68; 95% CI: 0.48–0.97).

Onitilo et al. performed a population-based cohort study that consisted of 9486 men with T2DM to examine the effects of glycemic control and use of anti-diabetic medication on the risks of breast, prostate, and colon cancer [52]. Use of metformin was associated with a reduced incidence of PCa (HR: 0.86; 95% CI: 0.72−1.03); however, other anti-diabetic medications had no similar association. No association with glycemic control was observed in patients with colon or breast cancer; however, the risk of PCa was significantly higher with better glycemic control, defined as hemoglobin A1c ≤ 7.0%. The results of this study implied that hyperinsulinemia, rather than hyperglycemia, is the major diabetes-related factor associated with increased risk of colon and breast cancers, while hyperglycemia is protective for PCa.

The Taiwanese National Health Insurance Reimbursement Database was used to investigate the effect of metformin use and PCa incidence [57]. This study included 186,212 metformin users and 209,269 metformin never-users. PCa incidence was calculated using a time-dependent approach, and HRs were estimated using Cox-regression analyses for ever-users, non-users, and subgroups for metformin exposure in tertiles of cumulative duration and cumulative dose. After adjustment with propensity-scoring, the HR for ever-users in relation to non-users was 0.467 (95% CI: 0.446–0.488). Moreover, the incidences decreased correspondingly to a longer cumulative duration and higher cumulative dose (*p* < 0.001 and *p* < 0.01, respectively) [57].

Not all population-based studies have observed a positive association between metformin use and reduced risk of PCa. Nordström et al. performed a population-based cohort study that included 185,667 men and showed that use of any anti-diabetic was not associated with risk of PCa (OR: 1.013; 95% CI: 0.816–1.257) [55]. Chen et al. used administrative databases of men with diabetes and analyzed the risk of PCa according to age groups and Chinese ethnicity [58]. Non-Chinese metformin users aged 50–59 had reduced risk (adjusted HR: 0.86; 0.74–1.00), but no association was observed in all Chinese men and non-Chinese men aged 60 years and above. The small sample size was a limitation of this study. Recently, 540 diabetic patients from the REDUCE study were utilized to investigate the relationships between metformin use and overall, low-grade, and high-grade PCa diagnosis. In this study, metformin use did not show any significant relationships with the diagnoses of overall PCa (OR: 1.19; *p* = 0.50), low-grade PCa (OR: 1.01; *p* = 0.96), or high-grade PCa (OR: 1.83; *p* = 0.19) [51].

A meta-analysis utilized data from 660,795 patients and 30 cohort studies related to metformin and PCa. Twelve cohort studies with PCa incidence as an endpoint were employed to analyze the effect of metformin. The results showed no association between use of metformin and PCa incidence (HR: 0.86; 95% CI: 0.55–1.34; *p* = 0.51). Subgroup analyses, according to the duration of use and cumulative dose, neither showed any association with the incidence of PCa [6].

The anti-cancer effects of anti-diabetics have not only been observed with metformin, but also with incretin-based therapies, such as GLP-1 receptor agonists and DPP-4 inhibitors [59]. Exendin-4, a GLP-1 receptor agonist, has been shown to reduce extracellular signal-regulated kinase (ERK)-mitogen-activated protein kinase (MAPK) phosphorylation in LNCap cells, which is a distinct mechanism from metformin which affects androgen receptor activation or apoptosis [59,60]. Exendin-4 diminished in vivo PCa growth when LNCap cells were implanted into athymic mice, and in turn, tumor expressions of P504S, Ki67, and phosphorylated ERK-MAPK were reduced [60]. A recent meta-analysis that included 52 studies that analyzed the association between glycemic control and the incidence of neoplasm in patients with T2DM showed that GLP-1 receptor agonists significantly reduced PCa incidence (HR: 0.66; 95% CI: 0.47–0.91; *p* = 0.01) [61].

In summary, numerous biological studies have proposed potential anti-cancer mechanisms of metformin in PCa. However, contemporary epidemiological studies and meta-analyses do not support the protective effect of metformin in PCa development. A larger and well-designed cohort study is warranted to elucidate the potential role of metformin in PCa risk.

## 5. Effect of Metformin on Recurrence-Free Survival

The efficacy of metformin use on recurrence-free survival (RFS) was investigated in various disease settings. Danzig MR et al. retrospectively investigated the efficacy of metformin on RFS in 767 diabetic patients who underwent RP and demonstrated that combined use of metformin and statins was associated with a higher incidence of RFS (HR: 0.19; 95% CI: 0.04–0.91) [62]. Taussky et al. evaluated the efficacy of metformin on RFS in localized PCa patients who received primary brachytherapy or RT with or without ADT. Metformin users showed a 50% reduction in RFS compared to diabetic non-metformin users and non-diabetics (HR: 0.5–0.6; *p* = 0.03–0.04) [63]. The substantial benefit of metformin use in patients treated with RT, however, was not consistent in all studies. Ranasinghe et al. reported no synergistic effect of metformin in patients who received RT (HR: 1.1; 95% CI: 0.7–1.6). Furthermore, there was no difference in RFS between patients who received high-dose (>1 g/day) and low-dose (≤1 g/day) metformin [64].

To address these inconsistent results, the effect of metformin on RFS has been a topic of several meta-analyses. Stopsack et al. utilized data from 9186 patients in nine retrospective cohort studies, five of which investigated RFS as the endpoint, and determined that metformin users exhibited a 21% improvement in RFS (HR: 0.79; 95% CI: 0.63–1.00) [65]. Xiao et al. extracted data from 177,490 individuals from 13 cohort studies, five of which investigated RFS as the endpoint. Random-effects modeling revealed metformin use to be significantly associated with improved RFS (HR: 0.74; 95% CI: 0.58–0.95) [66].

Not all meta-analyses have demonstrated a positive association between use of metformin and improved RFS following primary treatment. Hwang et al. used five RFS studies and showed no statistical significance for outcomes. The limited number of included studies and small sample size probably affected these results [67]. A recent meta-analysis included data from 660,795 patients in 30 cohort studies. Among these studies, eight were utilized to evaluate the effect of metformin on RFS according to type of previous treatment. In the overall group, metformin use was associated with improved RFS (HR: 0.60; 95% CI: 0.42–0.87). Metformin also improved RFS in the subgroup of patients treated with RT (HR: 0.41; 95% CI: 0.29–0.58). However, metformin did not affect RFS in the subgroup of patients who underwent RP (HR: 0.84; 95% CI: 0.53–1.33) or in patients with mixed treatments, including ADT and docetaxel chemotherapy (HR: 0.56; 95% CI: 0.25–1.26) [6].

## 6. Effect of Metformin on Overall Survival

To date, the efficacy of metformin use on OS outcomes has been investigated in eight studies (Table 2). While some studies reported that metformin improved OS [68,69,70], others showed no significant relationship [71,72,73,74,75].

Distinct results have been reported for the efficacy of metformin on OS in patients who had received external-beam RT or brachytherapy [70,72,75]. A single-center retrospective study utilized data from 2901 patients with localized PCa treated with external-beam RT and analyzed the benefit of metformin use (median dose of 500 mg, twice daily) on CRPC progression, OS, and CSS. The results showed improved outcomes in the metformin group compared with the diabetic non-metformin group with respect to CSS (HR: 5.15; 95% CI: 1.53–17.35), OS (HR: 2.25; 95% CI: 1.38–3.66), and progression to CRPC (aOR: 14.81; 95% CI: 1.83–119.9) [60]. However, Zaorsky et al. reported different findings [72]. This group retrospectively reviewed data from 3217 patients with T2DM and localized PCa undergoing definitive RT and showed that patients with T2DM who received metformin did not experience significantly improved OS and CSS outcomes compared to patients without T2DM. Taira et al. reported similar results in which metformin had no effect on OS outcome in 2298 patients who underwent permanent brachytherapy [75].

The efficacy of metformin use on OS was investigated in patients with metastatic CRPC treated with docetaxel chemotherapy. Mayer et al. utilized data from the Ontario Healthcare databases, and patients over 65 years of age were stratified into groups based on diabetic status and anti-diabetic medication use. Hyperglycemia, an adverse effect of docetaxel, has been suggested to reduce the efficacy of docetaxel in inducing PCa cell apoptosis. Therefore, the authors investigated whether metformin combined with docetaxel induced additional effects that included alleviation of the resistance caused by hyperglycemia in induction of PCa cell apoptosis [76]. However, no significant CSS (HR: 0.96; 95% CI: 0.79–1.16) or OS (HR: 0.94, 95% CI: 0.82–1.08) benefit from metformin use was demonstrated [73].

Joentausta et al. investigated the relationship between anti-diabetic medication use and OS utilizing data from 1314 patients who received RP at Tampere University Hospital. The results indicated that the risk of death increased with preoperative use of anti-diabetic medications (HR: 1.81; 95% CI: 1.03–3.19); however, there was no association for postoperative use of anti-diabetic medications or metformin on survival either by amount or intensity (defined daily dose/year: ≤120, 121–265, and ≥266) [74]. The results should be interpreted with caution since common limitations existed in these studies. First, the studies were retrospective in nature, and metformin was not a random variable. Therefore, bias may have existed regarding unmeasured confounding variables. Second, the non-exact timing and duration of anti-diabetic medication use may have induced “immortality time bias.” Third, the databases used in these studies were population-based. Although a randomized design was implemented to reduce selection bias, “healthy user bias” cannot be ruled out.

To overcome the limitations of individual studies, several meta-analyses have been performed to elucidate the efficacy of metformin use in improving OS outcomes. Stopsack et al. utilized data from 9186 patients included in nine retrospective cohort studies. As a result of analyzing six studies that investigated OS as an endpoint, metformin was associated with a superior OS outcome (HR: 0.88; 95% CI: 0.86–0.90) [65]. Xiao et al. extracted data from 177,490 individuals from 13 cohort studies, eight of which had OS as an endpoint. Again, metformin use was significantly associated with improved OS (HR: 0.79; 95% CI: 0.63–0.98) [66]. However, a meta-analysis performed by Hwang et al., which involved four studies that focused on OS, did not report statistical significance [67].

As with findings observed for RFS, the beneficial effect of metformin seems to be distinct according to type of previous definitive treatment. A recent meta-analysis included 660,795 patients from 30 cohort studies, 13 of which investigated OS. Metformin proved to be beneficial in terms of OS in the overall group (HR: 0.72; 95% CI: 0.59–0.88) and in the subgroups of patients who had received RT (HR: 0.44; 95% CI: 0.35–0.55), suggesting that metformin induced radiosensitizing effects. Moreover, metformin was beneficial in the subgroup of patients who received ADT (HR: 0.77; 95% CI: 0.74–0.81), suggesting that metformin exhibited therapeutic benefits that outweighed the detrimental effect of ADT on metabolic syndrome. However, the effect of metformin use was not significant in the subgroup of patients who received RP (HR: 0.61; 95% CI: 0.32–1.17) or in the subgroup of patients with mixed treatment (HR: 0.82; 95% CI: 0.53–1.25) [6].

## 7. Effect of Metformin on Cancer-Specific Survival

Six studies evaluated the efficacy of metformin use on CSS outcomes and reported different results in a variety of disease settings [69,70,71,72,73,78] (Table 2).

Two studies investigated CSS outcomes in patients who underwent external-beam RT or brachytherapy and observed conflicting results [70,72]. Spratt et al. reported a single-institutional retrospective study evaluating data from 2901 patients with localized PCa treated with external-beam RT and observed that metformin (median dose of 500 mg, twice daily) improved CSS (HR: 5.15; 95% CI: 1.53–17.35) [70]. In contrast, Zaorsky et al. retrospectively reviewed 3217 patients in an identical clinical setting and observed that patients receiving metformin exhibited comparable CSS with patients without T2DM [72].

Mayer et al. investigated patients who received docetaxel chemotherapy and failed to demonstrate any association between metformin use and CSS outcome. This study included patients with metastatic CRPC over 65 years of age who were treated with docetaxel chemotherapy. Patients were stratified according to diabetic status and anti-diabetic medication use. No significant effect of metformin with or without other anti-diabetic medications was observed for CSS (HR: 0.96; 95% CI: 0.79–1.16) [73].

To overcome the limitations of these cohort studies, several meta-analyses were performed. Stopsack et al. utilized data from 9186 patients in nine retrospective cohort studies, in which, as a result of analyzing four studies investigating CSS as an endpoint, there was no significant association between metformin use and CSS (HR: 0.76; 95% CI: 0.44–1.31) [65]. Xiao et al. extracted data from 177,490 individuals in 13 cohort studies, and six studies with CSS as the endpoint were used. Metformin use was significantly associated with improved CSS (HR: 0.76, 95% CI: 0.57–1.02) [66]. A recent meta-analysis included data from 660,795 patients and 30 cohort studies related to metformin and PCa. Among these cohort studies, seven with CSS as the endpoint showed that metformin was associated with improved CSS in the overall group (HR: 0.78; 95% CI: 0.64–0.94) [6]. Notably, this positive association was prominently observed in the subgroup of patients who received RT (HR: 0.18; 95% CI: 0.07–0.45), suggesting a radiosensitizing effect of metformin involving mitochondrial complex I inhibition and AMPK activation [77].

## 8. Clinical Trials

Clinical trials have been conducted to elucidate the relationship between metformin use and survival for patients with PCa (Table 3). Four phase II studies have been completed, and phase I and other phase II trials are ongoing.

The SAKK 08/09 trial (NCT01243385) was a single-arm, phase II trial that investigated the effect of metformin on progression-free survival and PSA doubling time in patients with CRPC. Metformin was administered continuously at 1000 mg twice daily consecutively for 4-week cycles. Among 44 enrolled patients, 36% were progression-free at week 12, and 9.1% were progression-free at week 24. Two patients had a confirmed ≥50% PSA decline. Prolongation of PSA doubling time following metformin administration was observed in 52.3% of patients. The results suggest that metformin provides an objective PSA response and may influence disease stabilization with a low toxicity profile [79].

The protective effect of metformin in preventing metabolic syndrome associated with ADT was investigated in a randomized, double-blind placebo-controlled phase II trial (NCT01620593). Non-diabetic patients beginning ADT for advanced PCa were randomized to either metformin (500 mg daily) or placebo group following castration. At weeks 12 and 28, no statistical differences were observed in regard to weight, waist circumference, and insulin level. At week 28, undetectable PSA was achieved in 50.0% of patients receiving metformin and in 53.3% of patients receiving placebo. These findings were not statistically significant. In contrast to the SAKK 08/09 trial, metformin conferred no impact on the risk of metabolic syndrome and had no anti-tumor effect [80].

The MetAb-Pro (NCT01677897) was a single-arm, phase II pilot study that investigated the impact of metformin combined with abiraterone in patients with metastatic CRPC progressing on abiraterone alone. Most patients experienced PSA progression, almost half showed radiographic progression, and one patient demonstrated symptomatic progression. Overall, metformin did not confer clinical disease progression-free benefit [82].

A single-arm, open-label, phase II trial investigated the effect of metformin on PSA response in patients with CRPC, and results are pending [81].

An additional clinical trial is investigating the efficacy of metformin as a monotherapy for patients planned for RT due to high-risk pathology following RP [83]. Another is investigating the combination of metformin and enzalutamide in patients with CRPC [84]. A third trial is investigating the efficacy of metformin as an adjunctive treatment to salvage RT in patients with biochemical failure following RP [85].

## 9. Conclusions

Contemporary standard treatments for advanced PCa are limited by drug resistance and toxicity. Therefore, new cellular targets and novel molecular therapeutic agents with favorable toxicity profiles are needed. Metformin exerts direct effects as a metabolic homeostasis regulator and indirect effects as an anti-proliferative and anti-carcinogenic agent. Considering the potential association between metabolic syndrome and PCa development and progression, metformin may be considered an adjuvant agent, both as a monotherapy and in combination with other therapies. Studies have reported conflicting results regarding the association of metformin use and the risk of PCa incidence and survival outcome. Ongoing trials are exploring the additional effects of metformin combined with androgen receptor axis-targeted agents in patients with CRPC and patients receiving salvage RT following RP. Overall, the relationship between use of metformin and PCa remains controversial. Additional studies are warranted to validate the clinical benefits and potential risks of metformin use. Studies to explore the optimal strategy to maximize the benefits of metformin’s intrinsic proprieties are necessary, as are studies of metformin’s pleiotropic effects, especially those related to reduction of serum glucose and insulin concentrations.

## Figures and Tables

**Figure 1 ijms-21-08540-f001:**
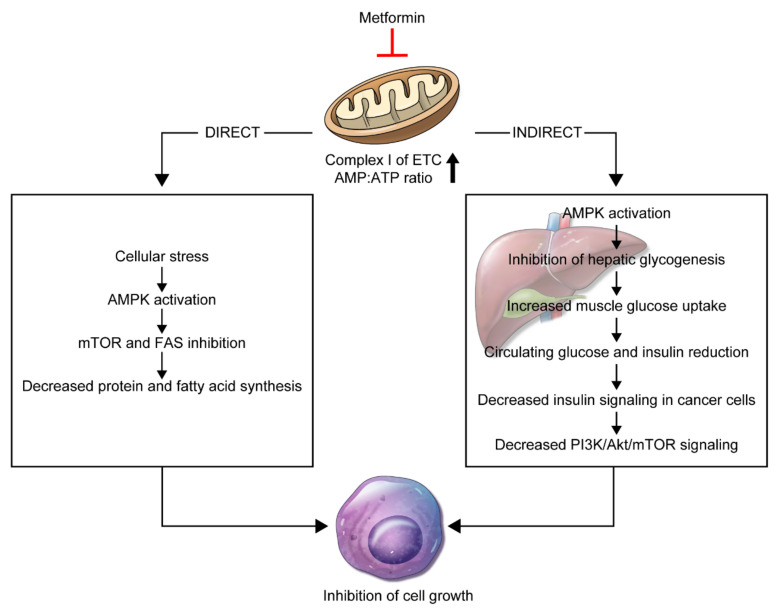
Schematic representation of the direct and indirect mechanistic anti-cancer effects of metformin. The reduction of mitochondrial oxidative phosphorylation is the fundamental action that ultimately inhibits the growth of host target cells.

**Table 1 ijms-21-08540-t001:** Epidemiological studies on the effect of metformin on prostate cancer incidence.

Author (Country)	Period	No. Cases/ Controls	Risk Estimates (95% CI)	Adjusted Variables
Preston (Denmark) [48]	1989–2011	12,226/122,260	Adjusted OR: 0.84 (0.74–0.96)	CCI, diabetic complications, marital status, use of PPI, statin, 5αRI
Haring (Finland) [49]	1996–1999	7681/1080	HR: 0.80 (0.67–0.96)	age, use of antihypertensives, cholesterol-lowering drugs, 5αRI, α-blockers, NSAIDs, aspirin
Ruiter (Netherlands) [50]	1998–2008	52,698/32,591	HR: 0.90 (0.88–0.91)	Age, sex, number of unique other drugs, number of hospitalizations, calendar time
Feng (REDUCE Study Group) [51]	2003–2009	194/205	Overall OR: 1.19 (0.72–1.99) Low-grade PCa OR: 1.01 (0.57–1.81)High-grade PCa OR: 1.83 (0.75–4.46)	Age, race, geographic region, PSA, prostate volume, digital rectal examination, BMI, family history of PCa, coronary artery disease, smoking, aspirin, NSAIDs, statin
Chen (Canada) [52]	1994–2012	35,829/44,172	Adjusted HR in non-Chinese men aged 50–59: 0.86 (0.74–1.00) aged 60–69: 1.00 (0.90–1.12) aged >70: 1.13 (0.99–1.29)	Socioeconomic class, use of other diabetes medications, dipeptidyl peptidase IV inhibitors, glucagon-like peptide 1 receptor agonists, thiazolidinediones, insulin
Wang (US) [53]	2003–2012	23,130/36,776	Non-Hispanic white HR: 0.91 (0.82–1.01) African American HR: 1.10 (0.94–1.27) Hispanics HR: 0.63 (0.49–0.80)	Age, race, CCI, BMI, LDL, HbA1c, PSA
Raval (US) [54]	2008–2009	948/1704	Unadjusted OR: 0.69 (0.49–0.95) Adjusted OR: 0.68 (0.48–0.97)	Age, race, marital status.
Nordstrom (Sweden) [55]	2007–2012	7678/177,989	Any PCa OR: 1.013 (0.816–1.257) High-grade PCaOR: 1.207 (0.936–1.557)	Age, log-transformed PSA level, PSA quotient, comorbidity, educational level, medication use
Onitilo (Australia) [56]	1995–2009	NA	HR: 0.86 (0.72−1.03)	HbA1c, glucose-lowering medication use (insulin, metformin, sulfonylurea), age, BMI, insurance status, comorbidities, smoking history, location of residence
Tseng (Taiwan) [57]	1998–2002	186,212/209,269	HR: 0.362 (0.345–0.380)	Age, PSA, comorbidities, obesity, other cancers

BMI: body mass index; CCI: Charlson Comorbidity Index; CI: confidence interval; 5αRI: 5alpha reductase inhibitor; HbA1c: hemoglobin A1c; HR: hazard ratio; LDL: low-density lipoprotein; NA: not applicable; NSAIDs: nonsteroidal anti-inflammatory drug; OR: odds ratio; PCa: prostate cancer; PSA: prostate-specific antigen; PPI: proton pump inhibitor.

**Table 2 ijms-21-08540-t002:** Studies on the effects of metformin use on cancer-specific mortality and overall mortality of prostate cancer.

Author (Country)	Population	Period	No. Cases/Controls	Risk Estimates (95% CI)	Adjusted Variables
Mayer (Canada) [73]	Patients receiving docetaxel chemotherapy	2005–2012	359/2473	CSM HR: 0.96, 95% CI: 0.79–1.16 OM HR: 0.94, 95% CI: 0.82–1.08	Age, use of statins and COX-2i, socio-economic status, urban/rural designation
Zaorsky (US) [72]	Patients receiving radiation therapy	1998–2013	251/2352	CSM sub-HR: 2.13 (0.90–5.08) OM sub-HR: 0.99 (0.65–1.52)	Age, comorbidities, PSA (log-transformed), Gleason score, T stage, ADT
Joentausta (Finland) [74]	Patients receiving radical prostatectomy	1995–2009	28/136	OM age-adjusted HR: 1.98 (0.86–4.53) multivariate-adjusted HR: 1.53 (0.57–4.08)	Age, use of 5αRI, preoperative PSA
Xu (US) [77]	Patients diagnosed with PCa, mixed therapy	1995–2010	3029/5910	OM (adjusted HR) Vanderbilt group: 1.04 (0.66–1.67) Mayo group: 0.69 (0.52–0.93)	Age, sex, race, BMI, tobacco use, insulin, cancer type, CCI
Randazzo (Switzerland) [69]	PSA-screened cohort, mixed therapy	1998–2003	150/4164	CSM OR: 27.94 (1.73–448.8) OM HR: 2.14 (1.19–3.87)	Age, PSA, family history, IPSS
Bensimon (UK) [71]	Patients newly diagnosed with non-metastatic PCa, mixed therapy	1998–2009	78/194	CSM overall RR: 1.09 (0.51–2.33) cumulative duration >938 days RR: 3.20 (1.00–10.24) OM post-diagnostic use of metformin RR: 0.79 (0.50–1.23)	Age, BMI, CCI, smoking, PSA, Gleason score, HbA1c excessive alcohol use, use of anti-diabetic agents (metformin, sulfonylureas, thiazolidinediones, insulins)
Spratt (Canada) [70]	Patients receiving radiation therapy	1992–2008	157/162	CSM (adjusted HR) non-diabetic group: 2.68 (0.85–8.44) * diabetic non-metformin group: 5.15 (1.53–17.35) * OS Non-diabetic group: 1.38 (0.90–2.11) * diabetic non-metformin group: 2.25 (1.38–3.66) *	Age, risk group, T stage, Gleason score, PSA, neoadjuvant ADT, BMI
Taira (US) [75]	Patients receiving interstitial brachytherapy	1995–2010	126/2172	OM HR: NA (p = 0.873)	Age, %positive biopsy, T stage, Gleason score, PSA, ADT duration, BMI, comorbidities
Margel (US) [78]	Patients newly diagnosed with PCa, mixed therapy	1997–2008	1251/2586	CSM HR: 0.76 (0.64–0.89)	Age, Johns Hopkins ACG Case-Mix System weighted sum of adjusted diagnostic groups, year of cohort entry, socioeconomic status, COX-2i, statins

ACG: adjusted clinical group; ADT: androgen-deprivation therapy; BMI: body mass index; CCI: Charlson Comorbidity Index; CI: confidence interval; COX-2i: cyclooxygenase-2 inhibitor; CRPC: castration-resistant prostate cancer; CSM: cancer-specific mortality; 5αRI: 5alpha reductase inhibitor; HbA1c: hemoglobin A1c; HR: hazard ratio; OM: overall mortality; OR: odds ratio; OS: overall survival; PCa: prostate cancer; PSA: prostate-specific antigen. * metformin group as reference.

**Table 3 ijms-21-08540-t003:** Clinical trials evaluating the efficacy of metformin in various disease settings.

Combination Agents	Clinical Phase	Identifier	Indication	Study End-Points	Status
Metformin monotherapy	II	NCT01243385 [79]	CRPC	PFS, OS, and safety	Completed
Metformin monotherapy	II	NCT01620593 [80]	Metastatic PCa	PFS, OS, PSA response, and safety	Completed
Metformin monotherapy	II	NCT01215032 [81]	CRPC	PSA response	Completed
Metformin + abiraterone	II	NCT01677897 [82]	Pre-chemotherapy CRPC progressing on abiraterone	PFS, OS	Completed
Metformin monotherapy	II	NCT02176161 [83]	Radical prostatectomy patients with high-risk pathology, prior RT with or without increasing PSA	PSADT	Ongoing
Metformin + enzalutamide	I	NCT02339168 [84]	CRPC	PSA response	Ongoing
Metformin + salvage RT	II	NCT02945813 [85]	Biochemical failure following radical prostatectomy	Time to progression, PFS, OS	Ongoing

CRPC: castration-resistant prostate cancer; OS: overall survival; PCa: prostate cancer; PFS: progression-free survival; PSA: prostate-specific antigen; PSADT: prostate-specific antigen doubling time; RT: radiation therapy.

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
