# Peer review of "Current Status and Application of Metformin for Prostate Cancer: A Comprehensive Review"

_ijms, 2020, doi:10.3390/ijms21228540_

Round 1

Reviewer 1 Report

In this paper, the Authors present a comprehensive review of the data on the possible effects of metformin in prostate cancer. However, the paragraph 3 (mechanisms of action of metformin) should be expanded to allow for a better understanding of clinical data. In particular, the cellular mechanism of action of metformin in the liver metabolism and at the mitochondrial level should be discussed. For example

Insulin/IGF:  PMID 28973479, 30165429

Liver: PMID 24847880

Hexokinase: PMID 25273809, 23797762

Caveolin/Cavin: PMID 32358634, 22038047, 23404184 , 23934189

Author Response

In this paper, the Authors present a comprehensive review of the data on the possible effects of metformin in prostate cancer. However, the paragraph 3 (mechanisms of action of metformin) should be expanded to allow for a better understanding of clinical data. In particular, the cellular mechanism of action of metformin in the liver metabolism and at the mitochondrial level should be discussed. For example

Insulin/IGF:  PMID 28973479, 30165429

Liver: PMID 24847880

Hexokinase: PMID 25273809, 23797762

Caveolin/Cavin: PMID 32358634, 22038047, 23404184 , 23934189

Response: Thank you for your thorough review. We agree that Section (3.2.) which describes the indirect effect of metformin was insufficient, and that the cellular mechanism of action of metformin in liver metabolism and at the mitochondrial level should have been discussed more thoroughly. According to the useful suggested references, we have added the information below to Section (3.2.).

The role of insulin in tumorigenesis is associated with the insulin receptor (IR), insulin-like growth factor receptor (IGF-R), and insulin-like growth factor (IGF). The insulin and IGF-1 receptors (IGF-1R) are homologous and often overexpressed in cancer cells. When insulin, IGF-1, or IGF-2 bind to IR, the signal is transmitted to the cells by autophosphorylation. Subsequently, the PI3K/AKT/mTOR signaling pathway is activated, leading to abnormal cell proliferation, inhibition of apoptosis, and carcinogenesis. According to several studies, IGF-1R is limited to caveolae, which are abundant in caveolin proteins that regulate vesicular transport, endocytosis, and cell signaling. Tyrosine phosphorylates caveolin 1, which is the most common caveolar protein, and has been shown to affect the anti-proliferative action of metformin.

Metformin also engages in glucose metabolism in cancer cells. Cancer cells exploit various mechanisms to produce energy, including increased metabolism, nutrient demand, and glucose consumption, known as the Warburg effect, to compensate for rapid cell growth and proliferation, primarily by glycolysis. Metformin suppresses glucose uptake in the tumor. The inhibitory action of metformin on glucose metabolism derives from the combination of glycolysis and its effect on the growth factor signaling pathway. Hexokinase (HK)-II is an enzyme that promotes glucose phosphorylation in the mitochondrial membrane and protects cells from cell death. Metformin inhibits HK-II, which catalyzes glucose phosphorylation, thereby reducing cellular energy availability and glucose uptake to promote cell death. In lung cancer, metformin has been shown to inhibit cellular glucose uptake and phosphorylation by impairing the enzyme functions of HK-I and HK-II in Calu-1 cells.

Reviewer 2 Report

The authors well summarized about current status and application of metformin for prostate cancer. Although the relationship between the use of metformin and prostate cancer remains controversial, metformin exerts direct effects as a metabolic homeostasis regulator and indirect effects as an anti-proliferative and anti-carcinogenic agent. I feel there are only several minor issues that should be revised.

  1. There is no description about the dose of metformin in the main text. You should mention about this point in the manuscript.
  2. Currently, anti-proliferative and anti-cancer effects on prostate cancer was reported in not only metformin but also GLP-1RA. I recommend you to describe the comparison of metformin and GLP-1RA.
  3. I feel English editing by a native speaker is required.

Author Response

The authors well summarized about current status and application of metformin for prostate cancer. Although the relationship between the use of metformin and prostate cancer remains controversial, metformin exerts direct effects as a metabolic homeostasis regulator and indirect effects as an anti-proliferative and anti-carcinogenic agent. I feel there are only several minor issues that should be revised.

Response: Thank you for your thorough review. We agree with all of the comments that were given, and have revised our paper according to the useful suggestions. Below are our answers to the specific comments/suggestions/queries.

1. There is no description about the dose of metformin in the main text. You should mention about this point in the manuscript.

Response 1: Thank you for this comment. We agree that information on metformin dose was not explicitly described. We have added information on the effects of dose-response relations in terms of intensity, duration, and cumulative dose whenever definable throughout the manuscript.

2. Currently, anti-proliferative and anti-cancer effects on prostate cancer was reported in not only metformin but also GLP-1RA. I recommend you to describe the comparison of metformin and GLP-1RA.

Response 2: We agree with the reviewer that the anti-proliferative and anti-cancer effects of GLP-1RA on prostate cancer are worth mentioning. Indeed, the anti-cancer effects of anti-diabetics have not only been observed with metformin, but also with incretin-based therapies, such as GLP-1 receptor agonists and DPP-4 inhibitors. Exendin-4, a GLP-1 receptor agonist, unlike metformin which affects androgen receptor activation or apoptosis, has been shown to reduce extracellular signal-regulated kinase (ERK)-mitogen-activated protein kinase (MAPK) phosphorylation in LNCap cells. Exendin-4 diminished in vivo prostate cancer growth when LNCap cells were implanted into athymic mice, and in turn, tumor expressions of P504S, Ki67, and phosphorylated ERK-MAPK were reduced. A recent meta-analysis that included 52 studies that analyzed the association between glycemic control and the incidence of neoplasm in patients with T2DM showed that GLP-1 receptor agonists significantly reduced prostate cancer incidence (HR 0.66; 95% CI, 0.47–0.91; p = 0.01). We have added this information to Section 4.

3. I feel English editing by a native speaker is required.

Response 3: Thank you for pointing this out. Although our first draft was reviewed by an English Editing Service, we agree that further corrections are necessary. We have re-submitted our revised version to an English Editing Service and have attached a Proof of Correction. Extensive revisions were made.